# Histological and metagenomic analysis of microbial communities in archaeological human bones

Damla Kaptan[1]*, Anne Cecilie Flemming Elvers[2], Anna Kjær Knudsen[2], Hannes Schroeder[2], Hege Ingjerd Hollund[1]

1 Museum of Archaeology, University of Stavanger, Stavanger, Norway, 2 Faculty of Health and Medical Sciences, Globe Institute, University of Copenhagen, Copenhagen, Denmark

* damla.kaptan@uis.no

## Abstract

Buried archaeological bones tend to be heavily degraded by microorganisms. This type of biodegradation was already identified in the 19th century and remains a subject of continuous investigation. However, the underlying processes are still not fully understood, and the organisms responsible for the decay have not been clearly identified. Technological advances in genetic sequencing now allow detailed study of the bone microbiome. And yet, identifying the species causing the observed bioerosion has proven challenging. Relatively few studies have combined the investigation of bone degradation by microscopy, so-called histotaphonomy, with metagenomic analyses. This study aims to bridge this gap. We utilize a large set of human bone samples from medieval cemeteries in south-western Norway. Detailed microscopic analyses have been carried out, showing diverse levels of preservation. The extent of bioerosion is correlated with the results from metagenomic analyses as well as environmental factors. Microbiome diversity is greater and more evenly distributed in well-preserved bones with limited bioerosion, particularly those recovered from burials beneath church floors, contrasting with outdoor cemeteries. Fungal taxa were detected in only a single sample in the metagenomic data despite histological evidence of fungal structures, and their role in bone bioerosion remains unclear. Our findings show that preservation state is strongly associated with microbiome composition. The most prevalent genus found was *Streptomyces,* supporting previous research suggesting that bacteria within this group could be involved in bone bioerosion.

## Introduction

Bone degradation by microorganisms occurs through enzymatic activities and demineralisation, a process known as bioerosion. After decades of research into bone diagenesis, the organisms causing deterioration of bone remain elusive. The

**Data availability statement:** All archaeological human remains sampled in this study are stored within the Museum of Archaeology, University of Stavanger. We have deposited all raw sequencing data (FASTQ files) in the European Nucleotide Archive (ENA) under the study accession PRJEB104929. The dataset is currently under controlled access and will be made publicly available upon publication of the manuscript, in accordance with journal and ENA data-sharing policies.

**Funding:** This work is part of an ongoing research project funded by the Research Council of Norway (project number 301877).

**Competing interests:** The authors have declared that no competing interests exist.

result of the deterioration, on the other hand, is well studied. Microscopy [1,2], porosity measures [3–7], and chemical analyses [8,9] show how microbial action alter the structure and chemistry of buried bones. Understanding bioerosion is crucial for comprehending bone decay due to microbial activity. Already in Child's article on bioerosion in archaeological bones from 1995, micro-organisms likely involved in bone decomposition were suggested based partially on the characteristics they are expected to have, and partially on cultivation experiments. However, his statement, that '...the organisms involved, have not yet been comprehensively defined' [10] still rings true today. Child (1995) also suggested that work on identification of the organisms involved should be combined with histological analyses of the bone samples [10]. Despite the recent interest in metagenomic analyses and environmental DNA, histological investigation of bone has rarely been combined with analyses of the bone microbiome [11–17]. This study aims to bridge this knowledge gap by investigating how the bone bioerosion and microbiome are correlated, and how the microbiome varies based on the preservation status and the environmental conditions of the biological remains. The ultimate goal is to provide new insights into the types of microorganisms that may be involved in the bone decay process. To address these questions, we analyzed bone samples from medieval cemeteries on the south-west coast of Norway, using light microscopy and scanning electron microscopy to describe and score the bioerosion patterns, and applied metagenome analysis to identify microorganisms. To our knowledge, this is the first time such detailed histo-taphonomic analysis of bioeroded bone samples from archaeological contexts are combined with metagenomic analyses of the bone microbiome.

Bioerosion of skeletal remains has been known since the 19th century, when Wedl made his first observations of tunnels in teeth and bone [18]. Later scholars have suggested that these were caused by fungi [19,20], however this has been contested and there is still debate on whether some of the tunnelling observed in bone can be connected to fungal activity or not [6,21,22]. The spongiform destructive foci frequently observed in archaeological bones is generally agreed to be caused by bacteria, being made up of remineralized bone matrix and fine sub-micron sized tunnels [6,21,23–25]. So far, however, despite decades of research and new, improved techniques, it has not been possible to confidently identify the microorganisms doing the actual bone bioeroding.

To be able to directly connect observed bioerosion in bone with the bone microbiome of the exact same sample, we combined microscopic analyses with metagenomic profiling of 83 medieval to post-medieval human bone samples of variable preservation level and assessed the result in relation to environmental conditions.

## Materials and methods

### Samples and sites

The National Committee for Research Ethics of Human Remains (Norway) approved the study of archaeological human remains with a written approval number 2020/131.

We analyzed bone samples from 83 individuals recovered from six cemeteries along the south-west coast of Norway, dating from the 11th-19th centuries (Fig 1, S1 Table).

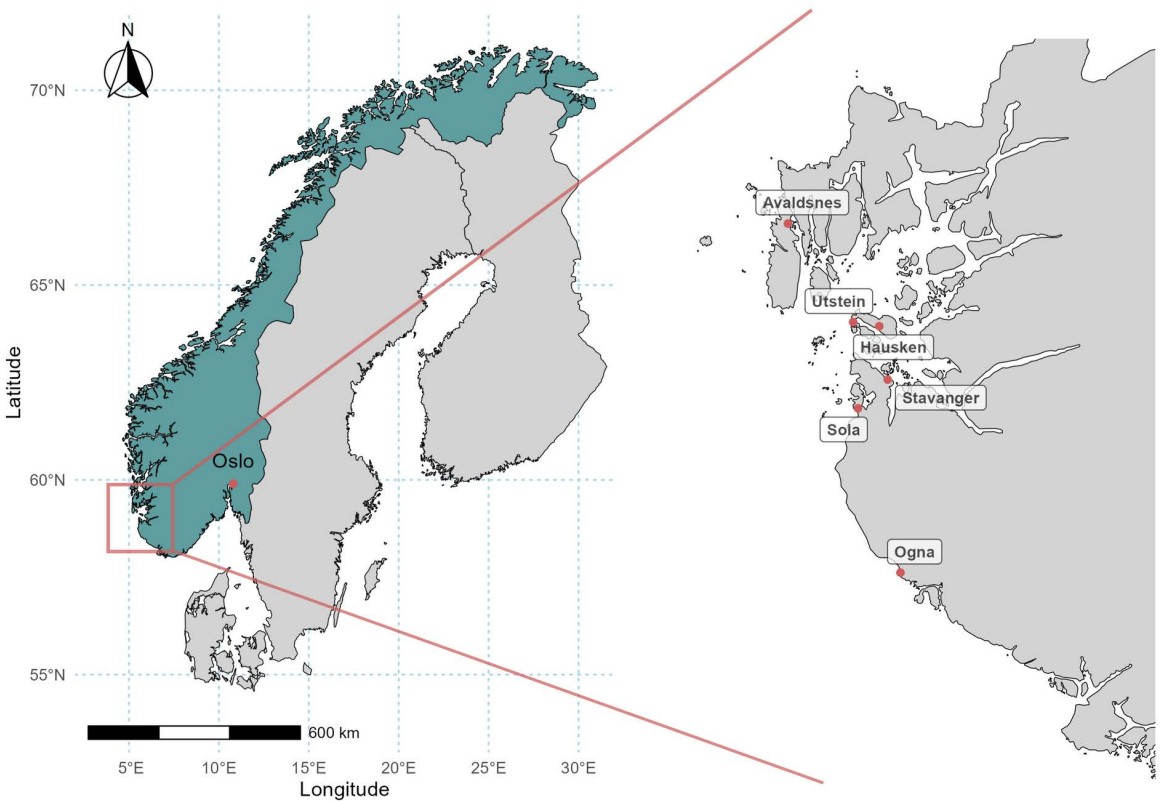

**Fig 1. Map of excavation sites.** Geographic location of the study area in Scandinavia and sampling sites in Rogaland, Norway. Country boundaries were obtained from Natural Earth (public domain). Administrative boundaries for Rogaland (NUTS-2 region NO04) were obtained from Eurostat GISCO. Maps were generated using R (ggplot2 and sf packages).

The majority of samples originate from the medieval town of Stavanger and its cathedral cemetery. Additional samples derive from the parish churches of Sola and Hausken, with a smaller number from Ogna parish church, Utstein Monastery, and the royal manor at Avaldsnes. The material was primarily obtained through rescue excavations and chance finds from the 1930s to 2022. In addition, 23 of the sampled individuals were uncovered during a research excavation in the old cemetery of Stavanger Cathedral in 2023. All individuals are recovered from soil burials, and in all cases where it could be confirmed, the remains were originally interred in wooden coffins. None were found in intact coffins, and most often only the iron rivets with bits of mineralised wood were preserved. Among the Stavanger cathedral burials, three individuals were originally interred in 17th-century brick-built grave chambers underneath the wood-covered floor of the nave and redeposited in the North graveyard in the 19th century (See S1 Appendix for more details).

Due to its robusticity and frequency, the femur is a commonly sampled skeletal element in bioarchaeology. In the present study, it was sampled whenever possible; when this was not feasible, the largest available long bone was selected. In cases where only skulls were present, cranial fragments were used (see Table 1, S1 Table, and S1 Appendix for further details). Some site- and context-specific subgroups contained very small sample sizes (n ≤ 3); analyses involving these subsets were therefore considered exploratory and should be interpreted with caution.

### Histological analysis: Bone bioerosion

Transverse sections (~5 mm) were cut from long bones using a Dremel rotary saw with a diamond blade. Samples were embedded in epoxy resin (Araldite 2020) under vacuum, polished to ~40 μm, and mounted on microscope slides without

**Table 1. Site and sample information summarised.**

| Site | Excavation year | Period | N | Environment/location |
|---|---|---|---|---|
| Stavanger East | 2022 | EM | 1 | Outdoor |
| Sola | 1986 | EM-PM | 17 | Outdoor |
| Ogna | 1994 | PM | 3 | Outdoor |
| Stavanger Byparken | 2005 | PM | 4 | Outdoor |
| Avaldsnes | 2020 | HM/LM | 2 | Outdoor |
| Stavanger Cathedral nave | 2021 | LM-PM | 12 | Indoor |
| Hausken | 2008 | LM-PM | 9 | Outdoor |
| Stavanger Cathedral choir | 1967 | VA/EM | 9 | Outdoor |
| Stavanger Cathedral North | 2023 | VA/EM – PM | 20 | Outdoor |
| Stavanger Cathedral North | 2023 | PM | 3 | Reburied |
| Utstein | 1930s-60s | HM-PM | 3 | Indoor |

Abbreviations for period: EM = early medieval, PM = post medieval, HM = high medieval, LM = Late medieval, VA = Viking Age.

coverslips. Thin sections were examined using light microscopy (Olympus BX51) in transmitted normal and polarized light using magnifications between x40 and x400. A subset of samples was further analyzed using a scanning electron microscope (Jeol JSM-IT800) equipped with Oxford Ultim Max 65 mm² EDS, operated either in low vacuum mode (no coating) or high vacuum with silver coating.

Bone preservation was quantified using the Oxford Histological Index (OHI) [26,27], which semi-quantifies bioerosion patterns (Table 2). OHI values (S1 Table) were analyzed in relation to temporal (period), spatial (site and indoor/outdoor), and excavation-year contexts. Multiple linear regression models were fitted using lm() in R to assess the combined effects of sample location (indoor vs. outdoor) and age (recent vs. old) on OHI values. "Old" refers to periods prior to the post-medieval, while "recent" refers to post-medieval periods. Both predictors (Location and Age) were included as categorical variables, with OHI treated as a continuous response variable. Model assumptions were evaluated using standard diagnostic plots, and statistical significance of model terms was assessed using ANOVA.

### Ancient DNA extraction and library preparation

A bone fragment for DNA analysis was collected immediately adjacent to the area used for histological assessment using a Dremel rotary saw fitted with a diamond blade. The excised bone samples (S1 Table) were subsequently powdered, with efforts made, where possible, to maximize the amount of cortical bone while avoiding surface contamination and trabecular bone. Ancient DNA (aDNA) extractions were performed in dedicated cleanroom facilities at the GeoGenetics Sequencing Core, University of Copenhagen.

Briefly, powdered bone samples underwent a short pre-digestion in 1 mL of incomplete digestion buffer (0.5 M EDTA, 30% N-Lauroylsarcosine) at 37 °C for 30 min. The buffer was then replaced with 1.8 mL of freshly prepared digestion buffer (0.5 M EDTA, 0.25 mg/ml Proteinase K, and 30% N-Lauroylsarcosine), followed by incubation at 37 °C for 48–72 h on a rocking platform.

aDNA extraction was performed using a protocol adapted for the Biomek i5 Automated Workstation (Beckman Coulter) [28], with the following modifications: For each sample, 150 µL of demineralised lysate were mixed with 1560 µL of binding buffer and 10 µL of magnetic silica beads, and incubated for 15 min with tip-mixing every 5 min. Pelleted beads were

**Table 2. Oxford Histological Index used for scoring the extent of bioerosion observed in a sample cross-section, following Millard [27].**

| Oxford Histological Index OHI | Approximate % of intact bone |
| --- | --- |
| 0 | <5 |
| 1 | <15 |
| 2 | <50 |
| 3 | >50 |
| 4 | >85 |
| 5 | >95 |

washed twice with 450 µL and 100 µL of 80% ethanol + 20% 10 mM Tris-HCl, respectively, and aDNA was eluted in 35 µL of 10 mM Tris-HCl + 0.05% Tween-20.

Double-stranded libraries were prepared [29], adapted to the Biomek i5 Automated Workstation (Beckman Coulter), and sequenced on an Illumina NovaSeq 6000 platform at the GeoGenetics Sequencing Core [30].

## Sequence processing and mapping

Raw sequencing reads were demultiplexed and processed using AdapterRemoval v2.3.3 [31]. Terminal Ns were removed, and reads were quality-trimmed (minimum base quality 2). Reads shorter than 30 bp after trimming were discarded. Overlapping paired-end reads were merged (collapsed) requiring a minimum overlap of 10 bp and allowing up to three mismatches. Both merged and unmerged reads passing filters were retained for downstream analyses.

## Taxonomic classification and authentication

Taxonomic classification was performed using aMeta [32], a pipeline designed for ancient metagenomic analysis that integrates *k-mer*–based classification via KrakenUniq [33] with competitive Lowest-Common-Ancestor alignment using MALT [34], including additional authentication and validation filters. For KrakenUniq, we employed an extended NCBI non-redundant (NT) database comprising microbial genomes (bacteria, viruses, archaea, fungi, and parasitic worms), the human genome, and complete eukaryotic genomes available at NCBI (downloaded from: [https://scilifelab.figshare.com/articles/online_resource/KrakenUniq_database_based_on_full_NCBI_NT_from_December_2020/20205504]). Reads were filtered to retain only those with at least 1,000 unique *k-mers* and 200 taxonomic reads. Bacteria with taxonomic reads exceeding 500 were processed further with MALT to assess coverage, post-mortem damage (PMD), and edit distance statistics.

## Microbial diversity and statistical analyses

For community-level analyses, taxonomically authenticated species-level assignments were collapsed to the genus level. For each sample, the number of detected taxa per genus was counted, resulting in a detection-based genus representation rather than a read abundance measure. Genera observed in only a single individual were excluded to reduce noise.

Alpha diversity was calculated using the Shannon index (diversity() function, vegan package), with group comparisons performed using Kruskal–Wallis tests and Dunn's post hoc tests (FSA package, Bonferroni-adjusted).

Bray–Curtis dissimilarities were calculated from the resulting unrarefied genus-level detection count matrix using the vegdist() function in the vegan package (v2.6.10) in R. Principal coordinate analysis (PCoA) of the Bray–Curtis dissimilarity matrix was performed using the cmdscale() function to visualize patterns of microbiome composition among samples. No rarefaction or read-count-based normalization was applied, as sequencing depth was broadly comparable across samples and read counts are known to be unstable in low-coverage ancient metagenomic datasets.

PERMANOVA analyses (adonis2() in vegan package) quantified the influence of environmental factors (age, location, preservation) on microbial community structure based on genus detection patterns. Homogeneity of dispersion was tested using betadisper() followed by ANOVA to ensure observed differences were due to composition rather than group variance.

All analyses were conducted in R (v4.3.2) within RStudio (2025.05.1 Build 513).

## Results

### Histological analyses: Bone bioerosion

Out of the collection of 83 bone samples, 70% displayed bioerosion with an OHI of 4 or less, and 43% were heavily bioeroded with an OHI of 0 or 1 (Fig 2). This means that 30% of the samples showed no bioerosion or such limited bioerosion that it did not affect the OHI score. Approximately 30% of the entire assemblage displayed intermediate levels of bioerosion, where larger parts of the bone remain unchanged (OHI of 2–4) (S1 Table). The well-preserved areas of bone in many of our samples, including the ones with OHI of 5, exhibited what Schotsman et al. (2024) termed 'onset of biodegradation (Fig 2 D and E) [35].

Fungal remains were observed in many thin sections, in the form of hyphae, fungal fruiting bodies, and spores. The fungal remains were mainly found within cracks and natural pores in the bone on or close to bone surfaces (S1 Fig). Two different types of hyphae were observed, often within the same sample, with different thickness, one roughly 1–2 µm across and the other 5–10 µm across. Both were reddish-brown color. The thicker hyphae were clearly branching, whereas the thinner hyphae were sometimes collected in dense bundles. Different shapes of fungal spores were also observed (S1 Fig). Possible biofilm was observed in the microscope. Light and electron images of examples can be seen in S2 Fig.

During the 2023 research excavation in Stavanger, three phases and groups of skeletal assemblages were apparent: 1) *In-situ* post-medieval burials (level I), 2) *in-situ* medieval burials (level II), and 3) re-deposited post-medieval coffins. The re-deposited remains, likely removed from the cathedral's burial chambers in the 19th century and reburied outside in their original coffins, display no bioerosion. However, only three of the nine individuals uncovered were sampled. Conversely, level II medieval burials were all heavily bioeroded. The *in-situ* post-medieval burials from level I also displayed extensive bioerosion, but more than half (6 out of 11) had an OHI of 2 or higher, and three samples were scored with an OHI of 5.

To assess the potential influence of period, site, and excavation year on bioerosion, we analyzed the OHI scores under varying conditions. An increasing trend in OHI was observed in more recent samples compared to those from older periods (Fig 3A). Further analysis of OHI values across different sites indicated that the highest OHI values were observed in indoor environments (Fig 3B). By categorizing the samples into "Recent" (post-medieval periods) and "Old" (earlier than post-medieval), a significant difference in the distribution of OHI values between recent and old samples was evident (Fig 3C). A detailed comparison between indoor and outdoor environments showed a significant difference, with indoor sites exhibiting higher OHI values (Fig 3D). However, the comparison of excavation year and OHI values did not reveal any specific relationship, apart from the observed pattern influenced by the indoor/outdoor effect (S3 Fig).

A multiple linear regression analysis was conducted to examine the effects of location (indoor vs. outdoor) and age (recent vs. old) on OHI values. The model was statistically significant, ($F_{(2, 80)}$ = 23.39, $p < 0.001$) and explained approximately 35% of the variance in OHI values (adjusted $R^2 = 0.353$). Outdoors samples showed significantly lower OHI values than indoor samples ($\beta = -2.26$, $p < 0.001$), whereas recent samples exhibited significantly higher OHI values than older samples ($\beta = 1.36$, $p < 0.001$).

### Metagenomic analyses: The bone microbiome

Extraction and library blanks, processed identically to the samples, showed no detectable microorganisms. KrakenUniq classification, however, identified 30 bacterial species and one fungal species spanning 16 genera in 83 individuals (S2 Table). Microorganisms exceeding 500 taxonomic reads in KrakenUniq were further evaluated using MALT. Only four species had more than 5,000 reads, each with a breadth of coverage exceeding 5% (S3 Table), with postmortem damage

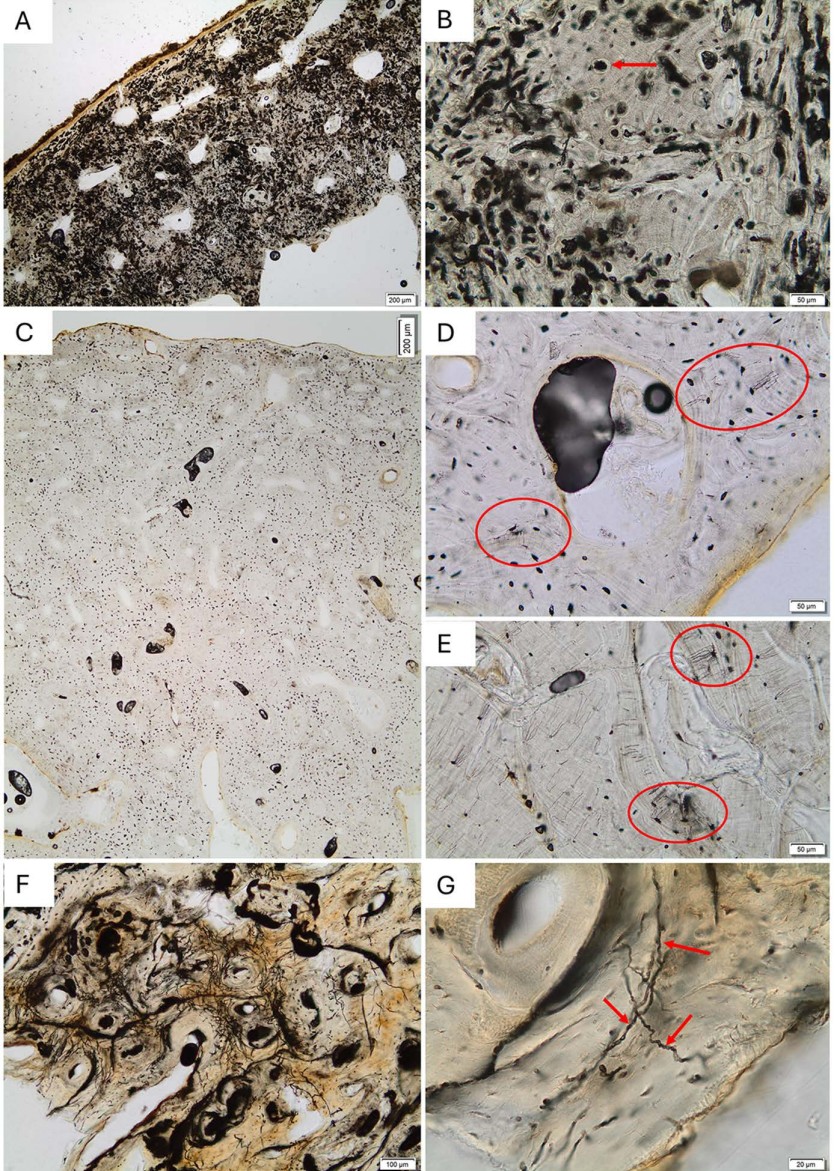

**Fig 2. Examples of bioerosion in the studied assemblage. (A)** Extensive bioerosion in sample S10294.17b; no original microstructure is visible. **(B)** Detail from the same sample as in **(A)**, showing characteristic destructive foci and small areas of well-preserved bone despite intense microbial attack. The arrow indicates a focal destruction surrounded by preserved bone. **(C)** Well-preserved sample (S14393.104) with no focal destruction observed. **(D)** Detail of the same sample, showing possible early-stage bioerosion (red ellipses), and **(E)** the same possible onset of biodegradation seen in sample S14393.80. **(F–G)** Wedl-type tunnelling observed in the mid-cortex of sample S14604.10 at two magnifications. Arrows in (G) indicate individual tunnels.

(PMD) profiles indicative of ancient origin. To enhance the reliability of taxonomic assignments, we restricted downstream analyses to the genus level and excluded any genera present in only a single individual, resulting in 13 bacterial genera retained across all samples. The most prevalent genus was *Streptomyces*, detected in 86% of individuals (Fig 4), followed by *Streptosporangium* and *Lysobacter*. In addition to bacterial genera, the fungal genus *Serpula* was detected in one sample from Stavanger Cathedral nave (S14393.72).

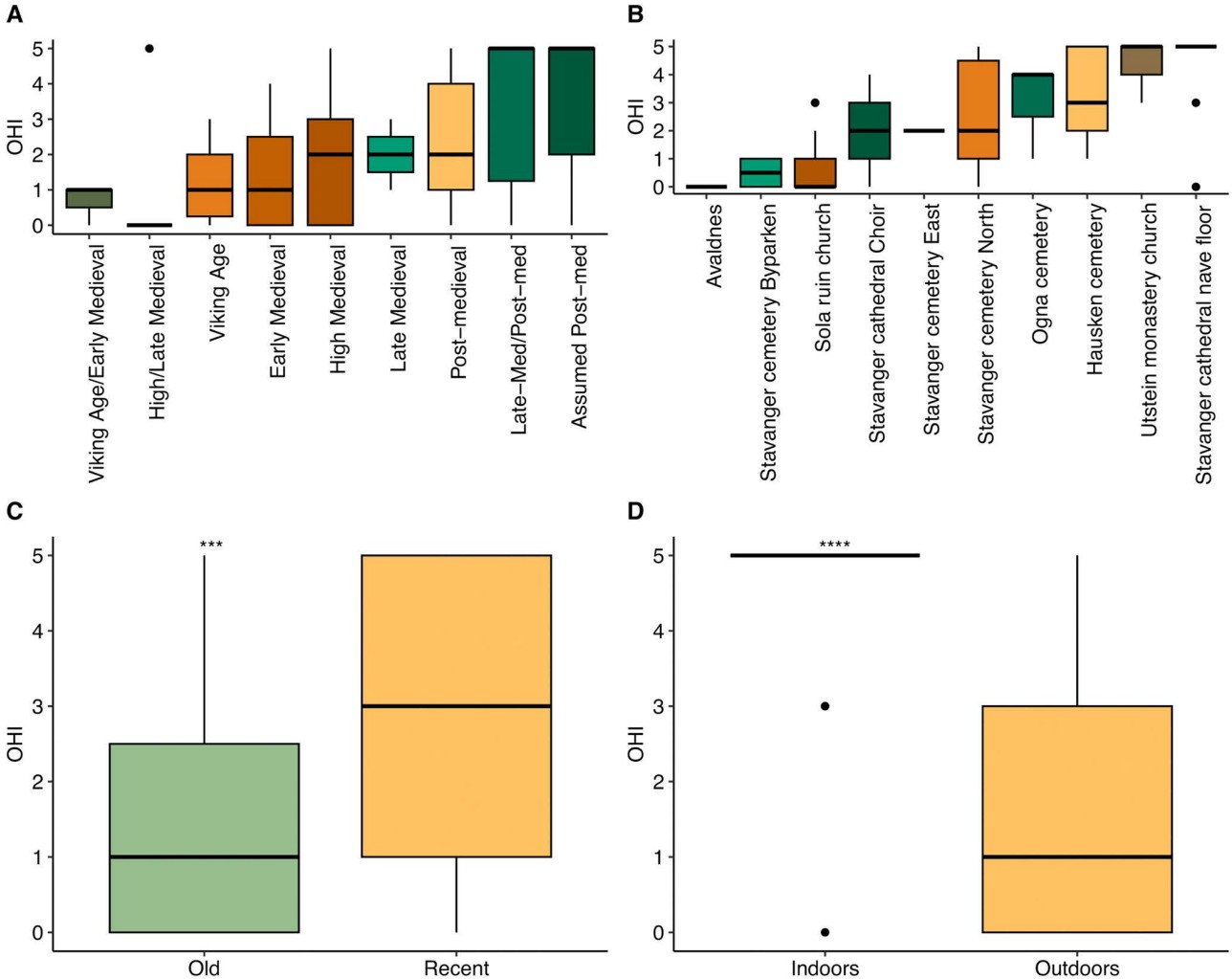

**Fig 3. Correlation between OHI values and different parameters. (A)** Sample period. **(B)** Excavation site. **(C)** Sample age (recent includes all post-medieval samples; old includes all earlier periods). **(D)** Sample location. Statistical significance was assessed using the Wilcoxon rank-sum test (*** $p < 0.001$; **** $p < 0.0001$).

Microbial alpha diversity, assessed using the Shannon index, was lower in "old" and/or "outdoor" samples compared to "recent" and/or "indoor" ones (Wilcoxon rank-sum test, $p < 0.005$; S4A–B Fig). Similarly, principal coordinate analysis (PCoA) based on Bray–Curtis dissimilarities showed that the distribution of samples along the first principal coordinate (PCoA1), explaining 54% of the variation, differed significantly by both sample age ($p = 0.03$) and burial location ($p = 0.003$) (Kruskal–Wallis test on PCoA1 scores; S4C–D Fig). No significant correlation was observed between excavation year and microbial diversity (S5 Fig).

To further investigate potential post-excavation effect, we compared well-preserved outdoor samples (OHI 4–5) grouped by excavation period: older excavations (Stavanger 1967, Ogna 1994, and Hausken 2008) and recent excavations (Stavanger 2023). Microbial diversity was assessed as the number of detected genera per sample. Although museum-stored samples from older excavations tended to show slightly higher microbial diversity than freshly excavated ones, the difference was not statistically significant (Wilcoxon rank-sum test, $p = 0.17$; S6 Fig).

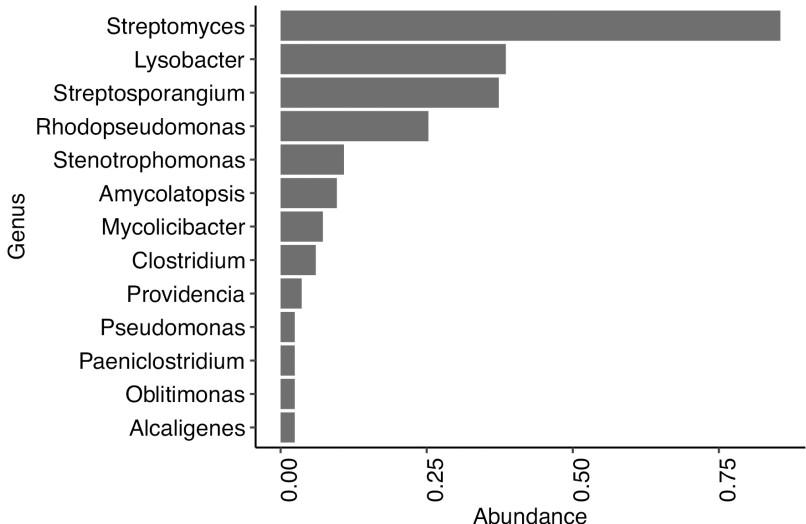

**Fig 4. Microbiome composition across all samples.** Only microbial genera observed in at least two samples are plotted.

PERMANOVA analyses on the full Bray–Curtis dissimilarity matrix confirmed significant effects of age (F = 3.23, R² = 0.049, p = 0.029) and location (F = 6.95, R² = 0.10, p = 0.001). A combined model including both factors explained 14% of the variance (F = 4.99, R² = 0.14, p = 0.001). The assumption of homogeneity of group dispersion was met for both Age (F = 1.81, p = 0.18) and Location (F = 0.09, p = 0.77), indicating that the observed differences in microbiome composition are not attributable to unequal within-group variance.

Observing that both the OHI and microbiome composition are influenced by burial environmental conditions, we investigated whether their correlation supports this pattern by assessing the impact of bioerosion on microbiome abundance. Samples were grouped by OHI scores, and genus-level microbial composition based on detection frequency was visualized (Fig 5). In samples with extensive bioerosion (OHI = 0, 1), *Streptosporangium* was the most dominant genus. In samples with moderate bioerosion (OHI = 3,4), *Lysobacter* predominated, while in well-preserved samples (OHI = 5), *Streptomyces* was the most abundant genus. Notably, samples with minimal bioerosion (OHI = 5) exhibited greater genus-level diversity compared to more degraded samples (OHI < 5).

To assess within-sample microbial diversity, Shannon diversity indices were calculated across samples grouped by preservation state (OHI levels) (S7A Fig). A Kruskal-Wallis test revealed a significant effect of preservation state on Shannon diversity (χ² = 15, df = 5, p = 0.01). Pairwise comparisons using Dunn's test (Bonferroni-adjusted) showed that samples with the highest preservation level (OHI 5) had significantly higher diversity than OHI 0 and OHI 1 samples (p = 0.05, and 0.03 respectively). To further explore these differences, the distribution of the PCoA1 derived from Bray-Curtis dissimilarities was examined. PCoA1 values varied significantly among OHI groups (Kruskal-Wallis χ² = 23.8, df = 5, p = 0.0002) (S7B Fig). Post hoc Dunn tests identified significant differences mainly between the lowest preservation states (OHI 0 and 1) and the highest (OHI 5) (*p* = 0.002, and 0.0007 respectively), indicating that better-preserved samples harbor distinct microbial communities. Overall, these results show that well-preserved bones tend to support richer and more evenly distributed microbial communities.

PERMANOVA analysis based on Bray–Curtis dissimilarities indicated a significant effect of preservation state on microbiome composition (F = 17.7, R² = 0.22, p = 0.001), suggesting that the evenness of microbial communities differed across preservation categories. A test for homogeneity of multivariate dispersion showed no significant differences among

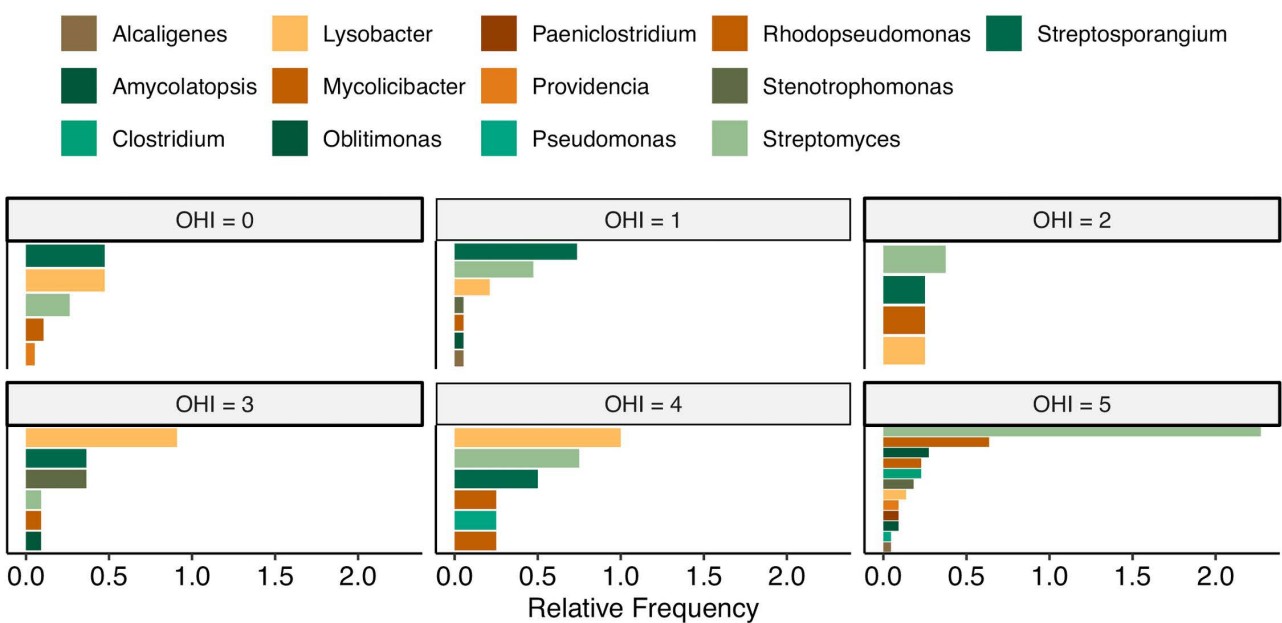

**Fig 5. Microbial composition of samples with different OHI scores.**

categories (F = 2.09, p = 0.079), indicating that the PERMANOVA result was not confounded by unequal within-group dispersion.

To visualize microbiome patterns across comparable samples varying in location and age, we compared genus-level detections between indoor nave burials and outdoor North burials at Stavanger Cathedral. The North burials were further subdivided into: North Level I (post-medieval, similar to the nave), North Level II (Medieval), and North redeposited burials. Indoor (nave) samples displayed a richer and more diverse microbiome, in contrast to North Level II burials, which showed only *Streptosporangium*. North Level I samples had more diversity than Level II but remained poorer than the Nave samples, while redeposited burials displayed an intermediate pattern between indoor and outdoor groups (Fig 6). These microbiome differences closely mirrored the OHI distribution, with the Nave exhibiting the highest preservation, North Level II the lowest, and a more diverse preservation pattern in the rest. Together, these findings demonstrate that preservation state is strongly associated with microbiome composition, reflecting and interacting with burial environment and postmortem conditions.

## Discussion

In this study, we utilized a large set of samples displaying different preservation levels, and combined histotaphonomic and metagenomic investigations. This diverse collection provides a suitable framework for exploring the variation in the metagenome, based on the presence, absence, and extent of bioerosion. The study represents a detailed analysis of human burials, encompassing various burial sites, periods, and excavation years, utilizing metagenomics. The aim is to understand bone histology and bioerosion in comparison with the metabiome.

### Environmental impacts on bone bioerosion and metagenome

Our primary interest lies in the degradation of biological remains in various burial environments. However, due to the inability to conduct environmental screening and control collection for older excavations, we lack the data required for an in-depth analysis of environmental factors affecting bioerosion. As a consequence, environmental effects are discussed

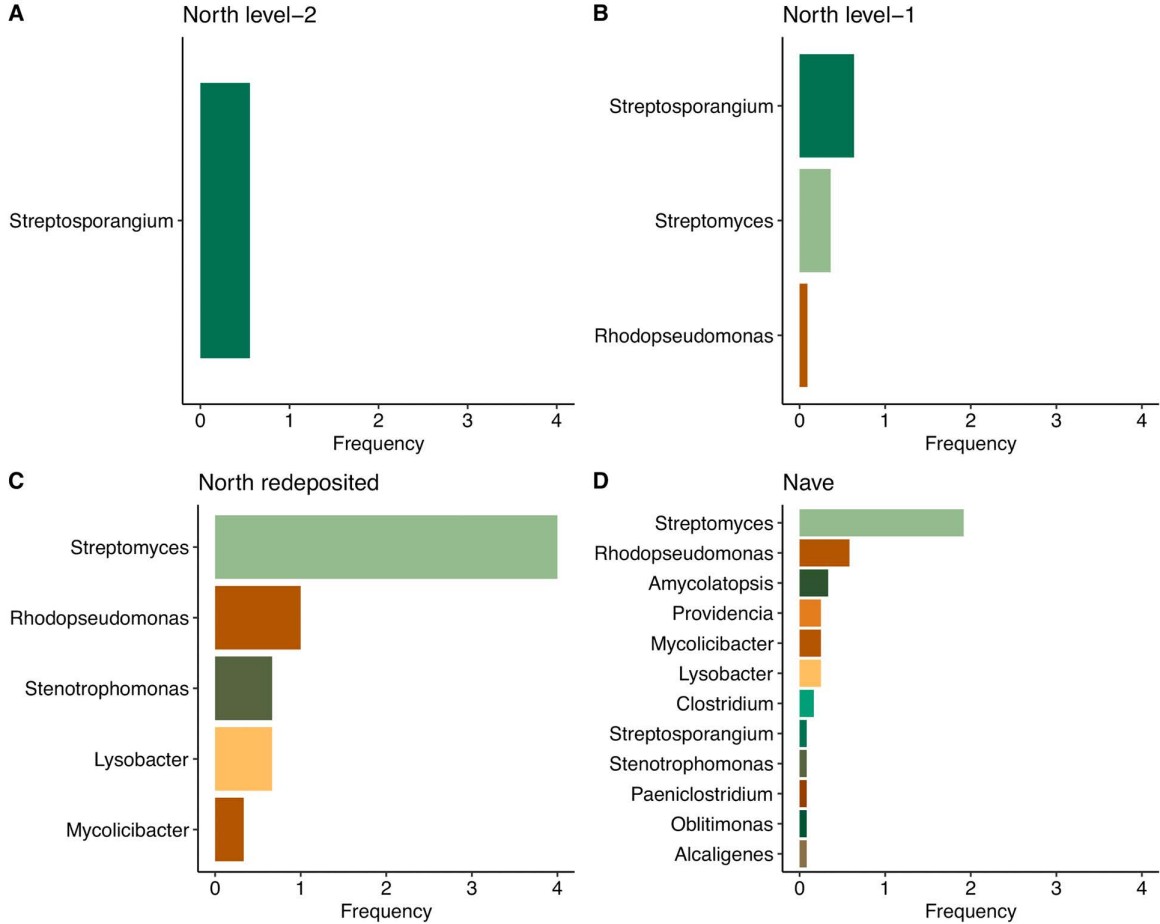

**Fig 6. Microbial composition across different burial contexts. (A)** North level 2 (n = 9). **(B)** North level 1 (n = 11). **(C)** North redeposited (n = 3), and **(D)** Nave burials (n = 12). For each genus, detections across samples were recorded, and detection frequency was calculated relative to the number of samples within each group.

here qualitatively and at the level of broad burial context (e.g., indoor vs outdoor). Nonetheless, the basic information obtained indicates that more recent biological material is better preserved in indoor burial environments, which is valuable for future conservation efforts.

Regression analyses identified burial location and age as significant predictors of histological preservation (OHI), explaining ~35% of variance. Both variables also influenced microbial diversity, confirmed by PERMANOVA, indicating that temporal and environmental factors independently shape bone microbiomes. Recent indoor samples displayed richer, more even microbial communities than older outdoor ones, despite the latter often showing greater bioerosion.

Both bone and the bone metagenome are generally better preserved in younger burials within an indoor environment, compared to old burials and/or outdoor cemeteries. Indoor burial conditions may limit microbial degradation of bone, which is consistent with expectations as the indoor materials are shielded from percolating rainwater and fluctuating burial conditions. Indeed, environmental monitoring of the cultural layers beneath the cathedral floor has shown that the sediments are relatively dry and exhibit a weakly alkaline pH [36], which is beneficial for the preservation of both bone [23,37,38] and DNA [39–41]. The alkaline pH, along with low moisture content, may have limited microbial access to bone proteins by reducing mineral dissolution, potentially contributing to the low levels of bioerosion [21,42]. The preservation of wood,

textiles, and plant remains in some graves [43], further attests to the protective nature of the indoor burial environment. Other environmental factors—such as infiltration of humic substances, organic decomposition byproducts, and metal ions from corroding artefacts, have been suggested as inhibitors of microbial bone decay [44,45] —although it is uncertain whether or not this would be substantial enough to explain the low bioerosion in the indoor cathedral remains. In contrast to the outdoor environment, where only iron rivets from coffins were preserved alongside the skeletons, numerous metal objects were found beneath the nave floor, both due to the rich coffins with metal furnishing, and due to pre-cathedral and construction phase activities [43].

Nine individuals in the Stavanger Cathedral North excavation were originally chamber burials from the 17th century, exhumed and reburied outside in the 19th century (S1 Appendix). Only three out of nine individuals discovered were analyzed. These represent a unique environmental and taphonomic history. When uncovered, all were in anatomical position, indicating that soft tissues were likely still intact at the time of reburial. Most likely, the chamber conditions were dry, leading to mummification by desiccation as reported in other above-ground 17th century graves [46]. Following reburial in soil, the remains became fully skeletonized, but the level of preservation is quite different from the in-situ burials. In two cases, hair was still preserved, and remnants of wooden coffins were present. All exhibited an OHI of 5 with no bioerosion. It is likely that the drainage trench into which the coffins were reburied, along with the coffin materials themselves, created a distinct microenvironment, differing from the rest of the excavation area. Comparisons of the microbiomes of these redeposited individuals revealed an intermediate pattern between the microbially diverse indoor (nave) samples and the low-diversity outdoor (north) burials (Fig 6). Further comparisons among the North burial subgroups reinforced this pattern of ongoing decay: North Level II burials, which were deeper and older, exhibited extremely limited microbial diversity, with only a single genus detected, whereas North Level I burials displayed slightly richer microbiomes. The detection of *Streptosporangium* as the sole genus in North Level II burials may indicate a unique group of microorganisms adapted to survive under nutrient-limited conditions in highly bioeroded bones. However, this hypothesis requires confirmation through further research on *Streptosporangium*.

Several sub-collections of the skeletal assemblage under study were excavated decades ago, the earliest in the 1930s (S1 Appendix). Previous studies suggest that excavated bones may contain active microorganisms including biofilm-forming microorganisms [12,47], and that the profile of the bone microbiome of museum-stored bones is different from the soil and from freshly excavated bones [17]. In our study, although the material stems from the same region and a relatively narrow time range, we do not have any directly comparable assemblages of bones from the same site but excavated at different times. Furthermore, parts of the collection were stored at different institutions, and we do not have detailed information on the storage environment throughout the history of the collections. Neither bone preservation nor microbiome composition showed any relationship with the year of excavation (S3 Fig), whereas strong site effects were observed. Heavily bioeroded bones display low microbial abundance (see discussion below) regardless of excavation year. However, comparisons restricted to well-preserved bones (OHI 4–5) from outdoor cemeteries revealed a pattern consistent with previous studies reporting higher microbial diversity in museum-stored bones, suggesting that microbial communities may continue to develop during storage and diverge from the original soil microbiomes [12,17,47].

## Microbial correlates of preservation

The complete aMeta pipeline resulted in only four microorganisms being confidently identified, as most taxa either lacked sufficient numbers of mapped reads or displayed irregular patterns of genome coverage for the species detected by KrakenUniq. A notable example is *Streptosporangium roseum*, which appeared in the majority of samples according to KrakenUniq but showed an inconsistent signal when re-evaluated with MALT. Although KrakenUniq reported numerous reads, MALT mapping revealed that these reads were confined to a few genomic regions and exhibited high mismatch rates, indicating that the assignment to the *S. roseum* reference genome is unreliable. A parsimonious explanation is that the detected reads derive from a related organism lacking a close representative in current reference databases—or

possibly from a yet-undescribed taxon—rather than from *S. roseum* itself. Given the ancient and poorly preserved nature of the analyzed material, low read counts were expected. Therefore, to ensure reliability, we report all detected taxa conservatively at the genus level rather than as confirmed species. Due to the instability of read counts across sequencing depths and library preparations in ancient metagenomic datasets, we deliberately focused on detection-based genus-level patterns rather than interpreting read numbers as proxies for microbial abundance.

Microbiome composition correlated strongly with OHI, with better-preserved bones (OHI 5) showing greater alpha diversity than degraded bones. This may reflect a continuum in which well-preserved bones retain more nutrients and structural integrity that support microbial colonization, whereas heavily degraded bones lack sufficient substrates. Taxonomic patterns support this interpretation. *Streptomyces*, a common soil genus known for its ability to use a wide-range of organic substrates [48], predominated in well-preserved bones, consistent with findings from Neanderthal and caribou remains [12,17]. In contrast, *Streptosporangium*, another actinobacterial genus, was more abundant in heavily bioeroded bones, while *Lysobacter* appeared mainly in moderately degraded ones. Notably, species from both *Streptomyces* and *Streptosporangium* genera have been shown to encode M09A-type collagenases—suggesting a potential for bone degradation [12]. Members of the genus *Lysobacter* have been shown through genomic and transcriptomic analyses to encode a diverse repertoire of extracellular lytic enzymes, including proteases with predicted collagenolytic potential, consistent with their role in soil suppressiveness and macromolecule degradation in other environments [49]. These distribution patterns may reflect stages of microbial succession or preservation-linked ecological niches, although their precise roles in bone diagenesis remain unresolved. Importantly, the observed association between microbial diversity and bone preservation does not allow discrimination between microorganisms acting as active degraders and those representing remnants of past colonization. The method used may capture DNA from inactive or dead microorganisms, as well as taxa introduced during excavation, handling, or long-term storage, and therefore represent a cumulative record of microbial presence rather than a snapshot of active communities. Consequently, the relationships described here should be interpreted as correlative rather than mechanistic.

The fungal genus *Serpula* was detected in only one sample in the KrakenUniq analysis. Although intriguing—possibly reflecting specific microenvironmental conditions—insufficient coverage prevents confident interpretation, and this finding should therefore be treated with caution. Nonetheless, fungal structures were observed histologically in this and several other samples. Similar shapes of hyphae and spores have been reported by Turner-Walker [21], on the surface of a medieval bone infected by fungi due to poor storage. Wedl-type tunnels, sometimes attributed to fungal activity [6,19,20,22], were also present in some of our specimens (S1 Fig). These fungal remains were generally confined to the outer bone surfaces, with only a few scattered microscopic fragments within the tissue. Most were likely removed during sample preparation, when outer layers were cleaned prior to aDNA extraction. It is also possible that fungi, as well as other contemporary microorganisms, are underrepresented in the metagenomic analysis due to the use of ancient DNA library protocols. Such libraries are generally optimized for fragmented and damaged molecules, which may bias detection toward older or more persistent DNA and potentially underrepresent contemporary microorganisms. Taken together, the histological and metagenomic observations indicate that fungi may be present, but the available evidence is insufficient to assess their role in bone bioerosion.

While our DNA-based analyses provide a comprehensive overview of microbial presence in archaeological bone, future studies incorporating RNA-based metatranscriptomic approaches could help capture the fraction of the microbiome that is potentially metabolically active. Transcriptomic data would allow the identification of genes involved in collagen and mineral degradation, providing functional insight into microbial contributions to bone bioerosion. Combining DNA and RNA profiling would therefore offer a more dynamic view of microbial activity and succession within the bone microenvironment, complementing the preservation and taxonomic patterns observed in this study.

Overall, our results indicate that bone microbiomes are associated with both preservation state and burial environment. The higher microbial diversity observed in well-preserved bones suggests that long-term preservation may coexist with microbial presence, underscoring the complex and dynamic nature of bone diagenesis.

## Conclusions

In this study, we made a direct comparison between the extent of bone bioerosion and composition of the bone microbiome, using a substantial and diverse collection of human bone samples from historic cemeteries and churches in south-western Norway. By analysing correlations between bone preservation, microbial communities, and environmental conditions, we found that higher microbial diversity was associated with well-preserved bones and indoor burials. Due to the nature of the sampled materials, we were unable to confidently assess the effects of post-excavation storage on microbial composition. Pinpointing the specific organisms involved in bioerosion remains challenging, particularly given the limited taxonomic resolution of ancient DNA data, which prevented species level identification. However, the results align with previous research suggesting the existence of a distinct "museum bone microbiome", and are consistent with the potential role of *Streptomyces* bacteria in bone degradation during soil burial. We found limited metagenomic evidence of fungi, and the available data do not allow us to clearly assess their role in bioerosion. Overall, our results contribute to a more nuanced understanding of bone bioerosion. Future research from other geographical regions and sites, particularly using well-documented and systematically sampled collections, will help to build on these findings and further clarify the mechanisms driving bone degradation.

## Declaration of generative AI and AI-assisted technologies in the manuscript preparation process

During the preparation of this work, the author(s) used ChatGPT to improve the grammar and clarity of the manuscript. After using this tool/service, the author(s) reviewed and edited the content as needed and take(s) full responsibility for the content of the published article.

## Supporting information

**S1 Appendix. Detailed description of excavation sites.**
(DOCX)

**S1 Fig. Microscopic and SEM Evidence of Fungal Structures.** (A) Fine hyphae and bundles of hyphae within the trabecular bone of S14393.83. (B) Detail of the fine hyphae and bundles in the same sample as in (A). (C) Thicker hyphae in sample S14393.72, clearly branching and segmented. The brown grains seen along the hyphae could be spores, or the cross-section of similar hyphae. (D) Fine hyphae (arrows) within two Haversian canals (blood channels) of sample S14393.118. (E) Fungal hyphae in a Haversian canal in S10294.18. The two large spheres are air bubbles in the embedding resin. (F) Spherical spores or sporangia within a Haversian canal in SZ20610. (G) A mass of almond-shaped spores filling a Haversian canal in SZ21164. (H) Lemon-shaped fungal spores in S14393.72. (I-K) SEM-images in backscatter mode of fine fungal hyphae, in pores of an unpolished bone sample of SZ20839. The diameter is roughly one μm and the hyphae have a hairy surface.
(TIF)

**S2 Fig. Examples of possible biofilm observed in the light and electron microscope.** (A) A grainy layer (black arrows) coating the surface of a Haversian canal and partially filling branching channels in SZ20610. Fungal hyphae are also present (arrows). (B) A grainy layer (black arrow) on the outer surface of A5133. Note the yellow etched surface area directly underneath (white arrow). (C) A grainy layered dome (black arrow) on the surface of a Haversian canal in SZ20711. (D) Similar grainy surface deposits were seen in the light microscope on sample STVA5132. Such a desposit (black arrow), and destructive foci in the bone underneath (red arrows), were studied further in the scanning electron microscope (E-G). (E) An SEM image of the same area as seen in the light micrsocope in (D), the arrows pointing to the same features. (F-G) SEM images of the surface deposit seen in (D) and (E) at higher magnification, in backscatter (F) and secondary electron mode (G). It has a dome-shape, consisting seemingly of a fibrous mass, and has low density. The

large cracks in the bone and between the deposit and the bone surface is likely caused by the sample preparation and/or the vacuum in the electron microscope. (H-J) Possible biofilm observed in sample SZ20839, within Haversiancanals (red arrows), in light microscope (H), and in SEM (I-J), where J shows a magnification of one part, revealing a porous structure and a low-density material. No chemical analyses were carried out, but it is clear from the back scatter image that there are no concentrations of dense materials such as manganese and iron thus the dark reddish-brown color seen in the light microscope is likely caused by an organic component.
(TIF)

**S3 Fig. Comparison of OHI values with excavation year.**
(TIF)

**S4 Fig. Diversity analyses of microbiome abundances.** (A) Shannon diversity between old and recent samples. (B) Shannon diversity between indoor and outdoor samples. (C) Distribution of the first principal coordinate (PCoA1) derived from Bray–Curtis dissimilarities between old and recent samples. (D) Distribution of the first principal coordinate (PCoA1) derived from Bray–Curtis dissimilarities between indoor and outdoor samples. "Old" refers to periods prior to the post-medieval, while "recent" refers to post-medieval periods. Statistical tests: Wilcoxon rank-sum test for Shannon diversity ($p < 0.005$) and Kruskal–Wallis test for PCoA1 differences (p_age = 0.03; p_location = 0.003).
(TIF)

**S5 Fig. Microbial diversity across excavation years.** (A) Shannon diversity (B) Distribution of the first principal coordinate (PCoA1) derived from Bray–Curtis dissimilarities.
(TIF)

**S6 Fig. Microbiome composition of outdoor samples.** Stavanger North Cemetery samples with only OHI scores 4 and 5, grouped into (A) Excavation Old (1967, 1994, 2008 excavations, n = 7) and (B) Excavation New (2023 excavation, n = 7).
(TIF)

**S7 Fig. Microbial diversity analyses between different preservation scores (OHI).** (A) Shannon diversity (B) Distribution of the first principal coordinate (PCoA1) derived from Bray–Curtis dissimilarities.
(TIF)

**S1 Table. Basic contextual information and histological results for all samples.**
(XLSX)

**S2 Table. KrakenUniq Abundance Matrix.**
(XLSX)

**S3 Table. MALT Sequence statistics.**
(XLSX)

## Acknowledgments

We are grateful to Sean Dexter Denham for conducting the excavation, providing archaeological context for the samples, and reviewing the text for language. We thank Bent Petersen for assistance with accessing and utilizing the servers at the GeoGenetics Sequencing Core. We thank Espen Undheim (University of Stavanger) for support with the SEM analyses, and Tom Gilbert (University of Copenhagen) for guidance regarding the genetics aspect of this study and for providing continued support for access to high-performance computing resources. We also thank Arda Sevkar (Hacettepe University) for support in metagenome analysis and for sharing his Bash scripts on aMeta.

## Author contributions

**Conceptualization:** Hannes Schroeder, Hege Ingjerd Hollund.

**Data curation:** Damla Kaptan, Hege Ingjerd Hollund.

**Formal analysis:** Damla Kaptan.

**Funding acquisition:** Hege Ingjerd Hollund.

**Investigation:** Damla Kaptan, Anne Cecilie Flemming Elvers, Anna Kjær Knudsen, Hannes Schroeder, Hege Ingjerd Hollund.

**Methodology:** Damla Kaptan, Anne Cecilie Flemming Elvers, Anna Kjær Knudsen, Hege Ingjerd Hollund.

**Project administration:** Hege Ingjerd Hollund.

**Resources:** Damla Kaptan, Hannes Schroeder, Hege Ingjerd Hollund.

**Software:** Damla Kaptan.

**Supervision:** Hege Ingjerd Hollund.

**Validation:** Damla Kaptan, Hannes Schroeder, Hege Ingjerd Hollund.

**Visualization:** Damla Kaptan, Hege Ingjerd Hollund.

**Writing – original draft:** Damla Kaptan, Hege Ingjerd Hollund.

**Writing – review & editing:** Damla Kaptan, Anne Cecilie Flemming Elvers, Anna Kjær Knudsen, Hannes Schroeder, Hege Ingjerd Hollund.

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
