## [Decision Letter · Decision Letter 0]

4 Feb 2026

PONE-D-25-63821Histological and Metagenomic Analysis of Microbial Communities in Archaeological Human BonesPLOS One

Dear Dr. Kaptan,

Thank you for submitting your manuscript to PLOS ONE. After careful consideration, we feel that it has merit but does not fully meet PLOS ONE’s publication criteria as it currently stands. Therefore, we invite you to submit a revised version of the manuscript that addresses the points raised during the review process.

If applicable, we recommend that you deposit your laboratory protocols in protocols.io to enhance the reproducibility of your results. Protocols.io assigns your protocol its own identifier (DOI) so that it can be cited independently in the future. For instructions see: https://journals.plos.org/plosone/s/submission-guidelines#loc-laboratory-protocols. Additionally, PLOS ONE offers an option for publishing peer-reviewed Lab Protocol articles, which describe protocols hosted on protocols.io. Read more information on sharing protocols at . Additionally, PLOS ONE offers an option for publishing peer-reviewed Lab Protocol articles, which describe protocols hosted on protocols.io. Read more information on sharing protocols at https://plos.org/protocols?utm_medium=editorial-email&utm_source=authorletters&utm_campaign=protocols..

We look forward to receiving your revised manuscript.

Kind regards,

Furqan A. Shah

Academic Editor

PLOS One

Reviewers' comments:

Reviewer's Responses to Questions

**Comments to the Author**

1. Is the manuscript technically sound, and do the data support the conclusions?

Reviewer #1: Yes

Reviewer #2: Yes

Reviewer #3: Yes

Reviewer #4: Yes

2. Has the statistical analysis been performed appropriately and rigorously? 

Reviewer #1: Yes

Reviewer #2: Yes

Reviewer #3: Yes

Reviewer #4: Yes

3. Have the authors made all data underlying the findings in their manuscript fully available?

Reviewer #1: Yes

Reviewer #2: Yes

Reviewer #3: Yes

Reviewer #4: No

4. Is the manuscript presented in an intelligible fashion and written in standard English?

Reviewer #1: Yes

Reviewer #2: Yes

Reviewer #3: Yes

Reviewer #4: Yes

5. Review Comments to the Author

Reviewer #1: This manuscript was a pleasure to read. The authors have presented their data well and their arguments are sound. I see no reason that this should not be published quickly with some small additions/modifications that I think will strengthen their paper. I am not best placed to comment on the details of the metagenomic analyses and will leave that to one of the other reviewers. Here my main focus has been on the histological analyses and their interpretation of that and the archaeological evidence.

Please read my comments on the annotated PDF file, there are rather a lot I'm afraid but these are "helpful" suggestions and comments rather than criticisms. I will reiterate a couple of the key ones here.

1) I never like it when I read about "enlarged canaliculi". The diameters of undegraded canaliculi in human bones are really at or beyond the limit of resolution for white light microscopy. For "enlarged canaliculi" I read "more visible canaliculi". The apparent increase in size is either caused by staining which both makes them more visible and alters the refractive index of the surrounding mineralised collagen matrix; or because canaliculi move out of the plane of focus and become visibly broader. In the authors Figure 2E it is the latter. Please see my comment in the caption for Figure 2.

2) There are some key points about the burial environments the skeletons were excavated from that appear in the discussion that really should be in the Materials section. I think they should be raised again in the Discussion where they can be related to the authors results. Perhaps a mention of soil water content and probable oxygen availability too if they have that data available to them.

3) The authors have this at the end of their discussion "Together, the histological and metagenomic evidence suggests that fungi do not play a central role in the bioerosion of bone and that the observed bioerosive patterns are predominantly bacterial in origin." I feel this is such an important conclusion that it should appear in both the Conclusions and the Abstract.

4) Also in the conclusion is this "Due to the nature of the sampled materials, we were unable to confidently assess the effects of post-excavation storage on microbial composition." The authors could add if they intend to tackle this issue in future work (or did I miss that?). Could the authors please add the paper by Provost et al. 2007. Freshly excavated fossil bones are best for amplification of ancient DNA. PNAS 104(3)

5) The authors make no comment about how their work relates to the "enteric hypothesis" for the origin of bone-degrading bacteria. This may be an intentional omission since they may not want readers to be distracted by such arguments - for which I don't blame them. Best to stick to the data of their own study.

Finally, well done. Nice work.

Reviewer #2: The work addresses an important and topical issue in bioarchaeology and taphonomy – the identification of microorganisms associated with the bioerosion of archaeological bones – and combines histotaphonomy (OHI) with metagenomic aDNA analysis.

More diversity in the microbiome was observed in better-preserved bones

Is it possible to determine whether the microbiome is a ‘remnant of colonisation’ rather than a degrading factor? Also, are heavily eroded bones simply too poor an environment to maintain diversity?

Some subgroups (e.g., redeposited burials, Utstein, Ogna) are very small. In my opinion, it should be noted in Methods/Discussion which comparisons are exploratory in nature, and conclusions based on n ≤ 3 should be formulated more cautiously.

It should be indicated whether the input data for Bray–Curtis was rarefied or normalised using another method.

Reviewer #3: Review of the manuscript “ Histological and Metagenomic Analysis of Microbial Communities in Archaeological Human Bones” submitted to Plos One. The manuscript is well-written and structured, and provides interesting insights in characterizing the biodegradation mechanisms affecting archaeological bones. I see a couple of points that show some weakness and I think that they are worth of a deeper discussion, improving the overall strength of the manuscript.

1) Environmental conditions likely play a major role in the microbiome onset; however these are only partially presented and discussed; I suggest to briefly summarize the environmental/climatic data available for the study cases (both current conditions and those encountered during burial, e.g. if climatic/environmental changes occurred in the area during the past centuries.

2) Linked to the previous point, I also suggest crosschecking if any correlation is observed between environmental conditions and OHI, similarly to what is observed in Fig. 3. Statistical analysis should provide evidence on the major factor(s) affecting bone integrity, and if the different environment can explain both the site- and age-specific diagenetic patterns.

3) Page 18, end of second paragraph. I suggest to improve the discussion on the limitations and potential bias occurring when using ancient DNA libraries to detect modern microorganisms.

Minor comments:

- Page 9, lines 3-5: the sentence is unclear, please rephrase

- Please mark figure 3 with A, B, C, D

Reviewer #4: The manuscript is clearly structured, presented in an intelligible manner, and written in standard English suitable for an international scientific audience.

The authors confirm that all data underlying the findings will be made fully available upon publication. Sequencing data and associated metadata will be deposited in an appropriate public repository, and accession numbers/DOIs will be provided at that stage. Additional relevant data will be included within the manuscript and its Supporting Information files. Therefore, all data underlying the findings in their manuscript are not fully available. Please make these data available before acceptance.

Once the authors have ensured that all data underlying the findings are made fully available, I strongly recommend the publication of manuscript PONE-D-25-63821 in PLOS ONE, as it presents a robust and innovative methodological framework by combining high-resolution histotaphonomic analyses with metagenomic profiling in a substantial set of medieval and post-medieval human bones. The study establishes quantitative correlations between the Oxford Histological Index (OHI), microbial diversity and environmental variables (burial age, indoor/outdoor context), using appropriate statistical approaches (linear regression, PERMANOVA, alpha and beta diversity analyses), and the results are consistent and well substantiated. The identification of distinct microbial profiles associated with different degrees of bioerosion, with a particular emphasis on the recurrent role of Streptomyces and Streptosporangium, makes a significant contribution to our understanding of bone degradation processes and to the ongoing discussion on the “museum bone microbiome”. The manuscript is clearly structured, well written, engages critically with both classical and recent literature, and offers relevant implications for taphonomy, conservation of osteological collections and ancient DNA studies, fully meeting the scope and methodological standards of rigour expected for PLOS ONE.

6. PLOS authors have the option to publish the peer review history of their article (what does this mean?). If published, this will include your full peer review and any attached files.). If published, this will include your full peer review and any attached files.

.

Reviewer #1: No

Reviewer #2: No

Reviewer #3: No

Reviewer #4: No

---

## [Author Response · Author response to Decision Letter 1]

20 Mar 2026

Corrections based on comments in the attached pdf:

• In line number 28, “Few” has changed to “Relatively few”

• In lines 46-47 the opening sentence has been changed to: “Bone degradation by microorganisms occurs through enzymatic activities and demineralisation, a process known as bioerosion.”

• Eriksen et al, 2020 reference is included in citation in the line number 59.

• Lines 65-67: Updated the sentence to “To our knowledge, this is the first time such detailed histotaphonomic analysis of bioeroded bone samples from archaeological contexts are combined with metagenomic analyses of the bone microbiome.” as the reviewer pointed out that a similar study has been carried out on experimentally buried bones.

• Line 69: Removed “Wedl” so that the sentence reads “Later scholars…”

• In line number 73 “mineral” has changed to “matrix”.

• Line 74-75: Rephrased sentence to: “So far, however, despite decades of research and new, improved techniques, it has not been possible to confidently identify the microorganisms doing the actual bone bioeroding”.

• In the lines 130-133 We have expanded the description of the linear regression analysis to clarify variable coding, model structure, and evaluation of model assumptions as: “Both predictors (Location and Age) were included as categorical variables, with OHI treated as a continuous response variable. Model assumptions were evaluated using standard diagnostic plots, and statistical significance of model terms was assessed using ANOVA.”

• Line 136: Number of the reference added instead of year, and “in transmitted normal light” removed from the table caption.

• Line 140-143: To expand upon how samples were collected, we added in the Methods section: “A bone fragment for DNA analysis was collected immediately adjacent to the area used for histological assessment using a Dremel rotary saw fitted with a diamond blade. The excised bone samples (S1 Table) were subsequently powdered, with efforts made, where possible, to maximize the amount of cortical bone while avoiding surface contamination and trabecular bone.”.

• In the line 208 “OHI” was corrected to “OHI score” as suggested.

• Please see the answer to comment #1 by reviewer-1.

• Comment on ‘which Fig 2: This section has been changed, see reply to comment#1 by reviewer-1.

• The original magnification of all images has not been reported, as all images contain a scale. However, a sentence has been added further on the microscopy, lines 121-122: “…using magnifications between x40 and x400”.

• Line 234: added “…reburied outside in their original coffins”.

• Question regarding evidence of coffin: We’ve added a sentence to the materials description: Lines 92-94: “All individuals are recovered from soil burials, and in all cases where it could be confirmed, the remains were originally interred in wooden coffins. None were found in intact coffins, and most often only the iron rivets with bits of mineralised wood were preserved.”

• In the line 298, “environment” was changed to “burial environment” to make the statement clear.

• In our dataset, microbial richness correlates with the level of preservation (OHI score), as discussed in the Discussion (lines 435–438). While the chemical/redox characteristics of the grave fill are indeed likely to influence microbial communities, we do not have the necessary screening data to evaluate the effect of redox conditions on microbial diversity in this study.

• Lines 94-97: Added some information on the burial chambers: “Among the Stavanger cathedral burials, three individuals were originally interred in 17th-century brick-built grave chambers underneath the wood-covered floor of the nave and redeposited in the North graveyard in the 19th century.”

• Because the detected organisms can only be confidently identified at the genus level, and because post-excavation inoculation of the bones cannot be excluded, it is not possible to reliably infer oxygen availability in these indoor structures based solely on the presence of these organisms.

• Lines 373-374: …low moisture and minimal water movement…, changed to “…low moisture content…”.

• Lines 376-380, about effects of metals and humics: We’ve modified the text to suggest these as possible inhibitors of microbial bone degradation, but that the extent of this effect in the cathedral environment is uncertain as: “Other environmental factors—such as infiltration of humic substances, organic decomposition byproducts, and metal ions from corroding artefacts, have been suggested as inhibitors of microbial bone decay [44,45] —although it is uncertain whether or not this would be substantial enough to explain the low bioerosion in the indoor cathedral remains.”.

• Lines 382-383: Information added on the indoor metal finds: “…both due to the rich coffins with metal furnishing, and due to pre-cathedral and construction phase activities”.

• Comment on hyphae and spores: A sentence and a reference added, lines 460-462: “Similar shapes of hyphae and spores have been reported by Turner-Walker [21], on the surface of a medieval bone infected by fungi due to poor storage.”

• We thank the reviewer for raising this point. We have now revised the text in lines 444-447 as “Members of the genus Lysobacter have been shown through genomic and transcriptomic analyses to encode a diverse repertoire of extracellular lytic enzymes, including proteases with predicted collagenolytic potential, consistent with their role in soil suppressiveness and macromolecule degradation in other environments [50].”.

• We added in the abstract lines 35-36: “No fungal species were detected genetically, and histological observations did not suggest any bone degradation by fungi.”.

• We added in conclusions lines 503-504: “We found no evidence of fungal degradation, suggesting that the observed bioerosion is mainly caused by bacteria.”.

Corrections based on comments of Reviewer#1:

1) I never like it when I read about "enlarged canaliculi". The diameters of undegraded canaliculi in human bones are really at or beyond the limit of resolution for white light microscopy. For "enlarged canaliculi" I read "more visible canaliculi". The apparent increase in size is either caused by staining which both makes them more visible and alters the refractive index of the surrounding mineralised collagen matrix; or because canaliculi move out of the plane of focus and become visibly broader. In the authors Figure 2E it is the latter. Please see my comment in the caption for Figure 2.

We agree that it’s an inaccurate term to use and perhaps it should be avoided altogether. We’ve been conservative in assigning this description to the samples, due to the issue that staining may make OCLs and canaliculi ‘more visible’ as the reviewer points out. However, we also see the point of the plane of focus. We’ve modified the text and changed two of the images, and instead of describing it as ‘enlarged’ reporting what Schotsman et al. recently termed ‘onset of degradation’, based on experimental work.

We’ve altered and expanded on the original text in line 210-212 in the result section:

• The well-preserved areas of bone in many of our samples, including the ones with OHI of 5, exhibited what Schotsman et al. (2024) termed ‘onset of biodegradation’ (Fig 2 A-E) [35].

We have also changed images and captions in Fig 2, lines 219-220:

• (D) Detail of the same sample, showing possible early-stage bioerosion (red circles), and (E) the same possible onset of biodegradation seen in sample S14393.80.

2) There are some key points about the burial environments the skeletons were excavated from that appear in the discussion that really should be in the Materials section. I think they should be raised again in the Discussion where they can be related to the authors results. Perhaps a mention of soil water content and probable oxygen availability too if they have that data available to them.

We do see the reviewer's point. However, we also see now that we forgot to refer to our supplementary document in this part of the materials section. This document provides detailed descriptions of the sites and excavations where the material is from. The skeletons stem from six different cemeteries and ten different excavation campaigns, and there is not enough space to detail this in the main text. Additionally, although Stavanger Cathedral is where we have the most detailed contextual information, and three interesting environments (indoor, outdoor and redeposited), our results summarise the observations across sites. We have, however made some additions and changes to the description of the Stavanger material in the supplementary document.

We’ve added the following sentences line 92-97:

• “All individuals are recovered from soil burials, and in all cases where it could be confirmed, the remains were originally interred in wooden coffins. None were found in intact coffins, and most often only the iron rivets with bits of mineralised wood were preserved.”

3) The authors have this at the end of their discussion "Together, the histological and metagenomic evidence suggests that fungi do not play a central role in the bioerosion of bone and that the observed bioerosive patterns are predominantly bacterial in origin." I feel this is such an important conclusion that it should appear in both the Conclusions and the Abstract.

We’ve added a sentence to the abstract, line 35-36:

• “No fungal species were detected genetically, and histological observations did not suggest any bone degradation by fungi.”

And to the conclusion, line 502-503:

• ” We found no evidence of fungal degradation, suggesting that the observed bioerosion is mainly caused by bacteria.”

4) Also, in the conclusion is this "Due to the nature of the sampled materials, we were unable to confidently assess the effects of post-excavation storage on microbial composition." The authors could add if they intend to tackle this issue in future work (or did I miss that?). Could the authors please add the paper by Provost et al. 2007. Freshly excavated fossil bones are best for amplification of ancient DNA. PNAS 104(3)

We have no immediate plan to tackle this issue, as our current dataset will not allow us. It would require the analysis of better-controlled and directly comparable skeletal assemblages, excavated at different points in time, or experiments. We are not assessing post-excavation effects on the preservation of human (endogenous) DNA. So, the Pruvost (2007) reference is not relevant here.

5) The authors make no comment about how their work relates to the "enteric hypothesis" for the origin of bone-degrading bacteria. This may be an intentional omission since they may not want readers to be distracted by such arguments - for which I don't blame them. Best to stick to the data of their own study.

This is indeed an intentional omission since we can only observe which microorganisms are there, but not say where they come from, whether it’s most likely soil or gut, or a mixture.

Finally, well done. Nice work.

Reviewer #2: The work addresses an important and topical issue in bioarchaeology and taphonomy – the identification of microorganisms associated with the bioerosion of archaeological bones – and combines histotaphonomy (OHI) with metagenomic aDNA analysis.

More diversity in the microbiome was observed in better-preserved bones

Is it possible to determine whether the microbiome is a ‘remnant of colonisation’ rather than a degrading factor? Also, are heavily eroded bones simply too poor an environment to maintain diversity?

We added in the lines 449-456 in discussion paragraph:

• “Importantly, the observed association between microbial diversity and bone preservation does not allow discrimination between microorganisms acting as active degraders and those representing remnants of past colonisation. Given the cross-sectional and DNA-based nature of the data, the detected microbiome may reflect historical colonization events rather than ongoing bioerosive activity. Alternatively, heavily bioeroded bones may represent a nutritionally and structurally impoverished environment that no longer supports diverse microbial communities. Consequently, the relationships described here should be interpreted as correlative rather than mechanistic.”

Some subgroups (e.g., redeposited burials, Utstein, Ogna) are very small. In my opinion, it should be noted in Methods/Discussion which comparisons are exploratory in nature, and conclusions based on n ≤ 3 should be formulated more cautiously.

It should be indicated whether the input data for Bray–Curtis was rarefied or normalised using another method.

We added in lines 108-110 the information as:

• “Some site- and context-specific subgroups contained very small sample sizes (n ≤ 3); analyses involving these subsets were therefore considered exploratory and should be interpreted with caution.”

We clarified that Bray–Curtis analyses were based on detection-based genus representation rather than read abundance, reflecting a conservative approach appropriate for low-coverage ancient metagenomic data. We updated:

Methods part in lines 183-186:

• “For community-level analyses, taxonomically authenticated species-level assignments were collapsed to the genus level. For each sample, the number of detected taxa per genus was counted, resulting in a detection-based genus representation rather than a read abundance measure. Genera observed in only a single individual were excluded to reduce noise.”

Methods lines 190-195 as:

• “Bray–Curtis dissimilarities were calculated from the resulting unrarefied genus-level detection count matrix using the vegdist() function in the vegan package (v2.6.10) in R. Principal coordinate analysis (PCoA) of the Bray–Curtis dissimilarity matrix was performed using the cmdscale() function to visualize patterns of microbiome composition among samples. No rarefaction or read-count-based normalization was applied, as sequencing depth was broadly comparable across samples and read counts are known to be unstable in low-coverage ancient metagenomic datasets”.

We further made adaptations to clarify the methodology, results and discussion parts in lines 301 as:

• “..based on detection frequency..”

Lines 339-342 (Figure caption):

“Fig 6. Microbial composition across different burial contexts.

(A) North level 2 (n=9). (B) North level 1 (n=11). (C) North redeposited (n=3), and (D) Nave burials (n=12). For each genus, detections across samples were recorded, and detection frequency was calculated relative to the number of samples within each group.”

And lines 431-434:

• “Given the instability of read counts across sequencing depths and library preparations in ancient metagenomic datasets, we deliberately focused on detection-based genus-level patterns rather than interpreting read numbers as proxies for microbial abundance.”

Reviewer #3: Review of the manuscript “ Histological and Metagenomic Analysis of Microbial Communities in Archaeological Human Bones” submitted to Plos One. The manuscript is well-written and structured, and provides interesting insights in characterizing the biodegradation mechanisms affecting archaeological bones. I see a couple of points that show some weakness and I think that they are worth of a deeper discussion, improving the overall strength of the manuscript.

1) Environmental conditions likely play a major role in the microbiome onset; however these are only partially presented and discussed; I suggest to briefly summarize the environmental/climatic data available for the study cases (both current conditions and those encountered during burial, e.g. if climatic/environmental changes occurred in the area during the past centuries.

While regional paleoclimate reconstructions for Northern Europe indicate broad trends such as the Medieval Warm Period (~950–1250 AD) and the transition toward cooler conditions later in the second millennium, there is no evidence of pronounced, short-term climatic extremes specific to Rogaland between 1000–1800 AD that could be linked directly to individual burial contexts. Moreover, detailed environmental records for the individual excavation sites are not available, preventing analysis of burial-specific climatic or redox influence

---

## [Decision Letter · Decision Letter 1]

6 Apr 2026

PONE-D-25-63821R1Histological and metagenomic analysis of microbial communities in archaeological human bonesPLOS One

Dear Dr. Kaptan,

Thank you for submitting your manuscript to PLOS ONE. After careful consideration, we feel that it has merit but does not fully meet PLOS ONE’s publication criteria as it currently stands. Therefore, we invite you to submit a revised version of the manuscript that addresses the points raised during the review process.

If applicable, we recommend that you deposit your laboratory protocols in protocols.io to enhance the reproducibility of your results. Protocols.io assigns your protocol its own identifier (DOI) so that it can be cited independently in the future. For instructions see: https://journals.plos.org/plosone/s/submission-guidelines#loc-laboratory-protocols. Additionally, PLOS ONE offers an option for publishing peer-reviewed Lab Protocol articles, which describe protocols hosted on protocols.io. Read more information on sharing protocols at . Additionally, PLOS ONE offers an option for publishing peer-reviewed Lab Protocol articles, which describe protocols hosted on protocols.io. Read more information on sharing protocols at https://plos.org/protocols?utm_medium=editorial-email&utm_source=authorletters&utm_campaign=protocols..

As the corresponding author, your ORCID iD is verified in the submission system and will appear in the published article. PLOS supports the use of ORCID, and we encourage all coauthors to register for an ORCID iD and use it as well. Please encourage your coauthors to verify their ORCID iD within the submission system before final acceptance, as unverified ORCID iDs will not appear in the published article. *Only* the individual author can complete the verification step; PLOS staff the individual author can complete the verification step; PLOS staff *cannot* verify ORCID iDs on behalf of authors.verify ORCID iDs on behalf of authors.

We look forward to receiving your revised manuscript.

Kind regards,

Furqan A. Shah

Academic Editor

PLOS One

Journal Requirements:

Reviewers' comments:

Reviewer's Responses to Questions

**Comments to the Author**

1. If the authors have adequately addressed your comments raised in a previous round of review and you feel that this manuscript is now acceptable for publication, you may indicate that here to bypass the “Comments to the Author” section, enter your conflict of interest statement in the “Confidential to Editor” section, and submit your "Accept" recommendation.

Reviewer #1: All comments have been addressed

Reviewer #2: All comments have been addressed

Reviewer #3: All comments have been addressed

Reviewer #4: All comments have been addressed

2. Is the manuscript technically sound, and do the data support the conclusions?

Reviewer #1: Yes

Reviewer #2: Partly

Reviewer #3: Yes

Reviewer #4: Yes

3. Has the statistical analysis been performed appropriately and rigorously? 

Reviewer #1: Yes

Reviewer #2: Yes

Reviewer #3: Yes

Reviewer #4: Yes

4. Have the authors made all data underlying the findings in their manuscript fully available?

Reviewer #1: Yes

Reviewer #2: Yes

Reviewer #3: Yes

Reviewer #4: Yes

5. Is the manuscript presented in an intelligible fashion and written in standard English?

Reviewer #1: Yes

Reviewer #2: Yes

Reviewer #3: Yes

Reviewer #4: Yes

6. Review Comments to the Author

Reviewer #1: The authors have done a good job of answering all of the reviewers' comments - either by making changes or by defending their original position. They are to be commended.

One nit-picking comment. The figure text for Figure 2 and Lines 219-220 should read "red ellipses". Yes, I am a pedant!

One last thing. Just after submitting my review of the first version of this paper I came across the attached illustration of Streptomyces sp. online. I think it bears a remarkable similarity to the authors' Figure 2F & G.

Reviewer #2: The revised version of the manuscript shows a marked improvement over the previous version. The authors have thoroughly addressed most of the reviewers’ comments, particularly by clarifying methodological aspects and expanding the discussion of the study’s limitations. The changes made have significantly improved the clarity and overall quality of the work.

The study is technically sound, with an adequate sample size and a properly applied approach combining histological and metagenomic analysis. The statistical analyses are appropriate, and the results are presented in a clear and organized manner.

Only minor issues regarding the interpretation of results remain. In particular, given the correlational nature of the data and the use of metagenomic methods, it would be advisable to consistently formulate conclusions in terms of associations rather than the direct functional involvement of specific microbial taxa. Similarly, the interpretation regarding the presence of fungi could be slightly clarified. I am referring to reflecting the discrepancies between microscopic observations and the results of metagenomic analyses.

However, these are editorial issues and can be easily corrected through minor changes in the Discussion and Conclusions section.

In summary, the manuscript is of high quality and represents a valuable contribution to research in this field. I believe that after minor revisions, it will be suitable for publication.

Reviewer #3: (No Response)

Reviewer #4: The authors have adequately addressed the comments raised in the previous round of review, and I now consider the manuscript acceptable for publication. It is technically sound, with the data fully supporting the conclusions. Following revision, I have verified that the statistical analyses were performed appropriately and rigorously. The authors have made all underlying data fully available. As a non-native English speaker, I am not best placed to assess this aspect comprehensively; nevertheless, the manuscript is presented intelligibly and in standard English.

In general, I recommend publication.

7. PLOS authors have the option to publish the peer review history of their article (what does this mean?). If published, this will include your full peer review and any attached files.). If published, this will include your full peer review and any attached files.

.

Reviewer #1: No

Reviewer #2: **Yes:** Maciej JaneczekMaciej Janeczek

Reviewer #3: No

Reviewer #4: No

---

## [Author Response · Author response to Decision Letter 2]

13 Apr 2026

Response to editor comments:

We have carefully reviewed the reviewer comments and confirm that no specific additional references were required. All cited literature has been selected based on its relevance to the study.

We have thoroughly reviewed the reference list to ensure that it is complete and accurate. No retracted articles are included. Minor updates and corrections to the reference list have been made where necessary and are reflected in the revised manuscript.

Response to Reviewer #1 comments:

We thank the reviewer for their positive evaluation of our manuscript and for their helpful comments.

The suggested correction has been implemented, and the figure text now reads “red ellipses” (Figure 2; Line 219 of the revised manuscript).

We also appreciate the reviewer’s observation regarding the similarity between the provided Streptomyces illustration and our Figure 2F & G. While this resemblance is interesting, we have refrained from making a direct taxonomic or functional attribution based solely on morphological similarity.

Response to Reviewer #2 comments:

We thank the reviewer for their positive and constructive evaluation of our manuscript. We appreciate their recognition of the improvements made in this revised version.

In response to the reviewer’s comments regarding the interpretation of the results, we have revised the Abstract, Discussion and Conclusions to ensure that our findings are consistently framed in terms of associations rather than direct functional involvement of specific microbial taxa. Statements implying causality have been carefully rephrased throughout the manuscript (e.g. Lines 25, 26, 347, 369, 374-375, 404-405, 432, 437, 438, 440, 441, 444, 445, 453-456, 481, 483, 491, 494, 497).

We have also clarified the interpretation of fungal presence by explicitly addressing the discrepancy between histological observations and metagenomic results. The revised text now highlights the potential influence of sampling, preservation, and methodological factors—such as surface cleaning and biases inherent to ancient DNA library preparation—on the detection of fungal DNA (e.g. Lines 35-37, 460, 467-473, 498-499).

These revisions have been incorporated into the Discussion and Conclusion sections of the manuscript. We hope that these changes adequately address the reviewer’s concerns.

---

## [Editor Report · Decision Letter 2]

14 Apr 2026

Histological and metagenomic analysis of microbial communities in archaeological human bones

PONE-D-25-63821R2

Dear Dr. Kaptan,

We’re pleased to inform you that your manuscript has been judged scientifically suitable for publication and will be formally accepted for publication once it meets all outstanding technical requirements.

An invoice will be generated when your article is formally accepted. Please note, if your institution has a publishing partnership with PLOS and your article meets the relevant criteria, all or part of your publication costs will be covered. Please make sure your user information is up-to-date by logging into Editorial Manager at Editorial Manager® and clicking the ‘Update My Information' link at the top of the page. For questions related to billing, please contact  and clicking the ‘Update My Information' link at the top of the page. For questions related to billing, please contact billing support..

Kind regards,

Furqan A. Shah

Academic Editor

PLOS One
---

## [Editor Report · Acceptance letter]

PONE-D-25-63821R2

PLOS One

Dear Dr. Kaptan,

I'm pleased to inform you that your manuscript has been deemed suitable for publication in PLOS One. Congratulations! Your manuscript is now being handed over to our production team.

Kind regards,

on behalf of

Dr. Furqan A. Shah

Academic Editor

PLOS One